# A Comprehensive Evaluation Method for Industrial Sewage Treatment Projects Based on the Improved Entropy-TOPSIS

**Xin Yu [1], Sid Suntrayuth [1] and Jiafu Su [2,*]**

[1] International College, National Institute of Development Administration, Bangkok 10240, Thailand; yuxin1007@163.com (X.Y.); sidsuntrayuth@hotmail.com (S.S.)
[2] National Research Base of Intelligent Manufacturing Service, Chongqing Technology and Business University, Chongqing 400067, China
* Correspondence: jeff.su@cqu.edu.cn; Tel.: +86-153-2032-3481

**Abstract:** Sewage treatment and reuse have always been hot issues in both the business and academic communities in all nations around the world. In order to solve the difficulties in accurate quantization and objective evaluation of industrial sewage treatment projects, this paper proposed a comprehensive industrial sewage treatment project evaluation method based on the improved entropy–TOPSIS method. First, this paper constructed an evaluation indicator system for sewage treatment projects from the four aspects of environmental performance, economic performance, managerial performance and social performance. Second, it made a modification to the experts' experience-based grading using the entropy weight method and determined the weight of the indicators in a more objective and more accurate manner. Third, this work improved the traditional TOPSIS method and simplified the calculations with regard to the traditional TOPSIS-based comprehensive evaluation. Finally, by taking the example of evaluating industrial sewage treatment projects of the China Water Affairs Group in Q city (ChongQing), China, this paper verified the feasibility and practicability of the proposed comprehensive industrial sewage treatment project evaluation system and method.

**Keywords:** industrial sewage treatment projects; environmental protection; comprehensive evaluation; entropy weight method; TOPSIS method

---

## 1. Introduction

In the past one hundred years, resource utilization and environmental protection have been a deep concern of all nations in the world. Most resources that are vital to human survival are limited, including land resources, mineral resources, coal, petroleum and natural gas, etc., and water in particular. Although most of the earth's surface is covered by seawater, fresh water is scarce around the globe, as it accounts for only 6% of total water resources. Furthermore, water resources are unevenly distributed in different regions. At present, the total volume of global water resources is about 1.4 billion cubic kilometers, and that of global freshwater resources is about 35 million cubic kilometers, wherein industrial and agricultural water consumption account for 25% and 70%, respectively. North America is endowed with abundant freshwater resources, which account for over 10% of global freshwater resources. The glaciers and snow on the vast prairies are the main sources of its fresh water. However, research hase shown that the volume of freshwater resources has been declining [1,2]. Some areas are even facing a scarce water supply. Due to global warming, Australia is also suffering from a certain degree of drought. As water resources become scarcer, the government has been taking positive measures. Asia also faces water pollution and drought.

In India in particular, heavy discharge of wastewater has polluted underground water and caused severe harm to people's health.

As global water resources are becoming scarcer, sewage has been recognized as a source of water. As such, sewage treatment and reuse have become an important means to solve water shortages. In developed countries, much attention and investment has been drawn to sewage treatment. The investment in sewage treatment in America, Japan and the UK, etc., accounted for 0.29%–0.55% and 0.53%–0.88% of the gross national product, respectively, in the 1970s and 1980s. However, in all countries around the world, due to both internal and external reasons for the establishment of a large quantity of sewage treatment projects and the complexity of such projects, the study on the evaluation of sewage treatment projects has relatively fallen behind, thus causing many problems during the operation of sewage treatment projects. The actual benefit of investment is lower than the expected benefit; some projects even face problems such as long-term failure to meet the design capacity, lower resource utilization rates, unreasonable structures of professional talents, excessive environmental pollution and poor capacity for repayment of loans. These problems are caused by inappropriate evaluation, selection and investment decisions of sewage treatment projects. However, the evaluation and selection of sewage treatment projects is quite a complex comprehensive decision-making problem that involves a wide range of factors, such as technology, the economy and the environment [3].

Therefore, an in-depth, meticulous and scientific analysis of these factors is required to provide correct guidance on the selection, design and implementation of industrial sewage treatment projects. In this paper, in order to ensure fast and healthy development of industrial sewage treatment, reasonable evaluation, the selection of and investment in sewage treatment projects and to maximize the conservation of social resources and energy, this paper will carry out an in-depth study on the evaluation and selection of industrial sewage treatment projects using systematic engineering methods such as the entropy weight method and TOPSIS (Technique for Order Preference by Similarity to an Ideal Solution) method, thus providing evaluation method support and decision-making support to decision makers in selecting industrial sewage treatment projects that are more suitable to actual conditions in a more scientific and more reasonable way.

## 2. Related Works

Extensive studies have been carried out on the comprehensive evaluation of projects in developed countries like the UK, America, Germany and France, and a lot of theoretical achievements on comprehensive evaluation have been made [4–6]. The most used comprehensive evaluation methods include the qualitative evaluation method, multiattribute decision-making method, operational research method, systematic engineering method, fuzzy mathematical method and intelligent evaluation method, etc. [7–10]. In recent years, with regard to the evaluation of sewage treatment projects, several attempts have been made to address industrial sewage treatment project evaluation and selection problems using various multicriteria decision-making methods and involving stakeholders' or experts' opinions [11–13]. Specifically, Jamwal et al. [14] adopted the Delphi method to evaluate the effectiveness of different technical projects for sewage treatment. Xia et al. [15] established an evaluation indicator system for sewage treatment projects in rural areas of China that took into account technical and economic factors, effectiveness and appropriateness and adopted a fuzzy advantages and disadvantages coefficient method for comprehensive evaluation of sewage treatment projects. Kalbar et al. [16] applied the TOPSIS method to evaluate three sewage treatment technologies, the activated sludge process, sequencing batch reactors and membrane bioreactors, by considering seven criteria. Gao and Zhang [17] used a three-phase data envelopment analysis (DEA) model for measurement and evaluation of urban sewage treatment projects before and after elimination of the influence of environmental factors and random errors. Peng et al. [18] applied set pair theory to construct a model for comprehensive evaluation of operational results of sewage treatment works and developed a new comprehensive evaluation method for sewage treatment operations plans based on

an improved analytic hierarchy process. Grimsey and Lewis [19] adopted a case study and expert workshop to make an in-depth study of risk evaluation of infrastructure projects for sewage treatment. Zhang et al. [20] carried out a comprehensive evaluation of full life-cycle costs and benefits with regard to sewage treatment and recycling projects using a full life-cycle evaluation method. Zhang et al. [21] and Yang et al. [22] discussed the risk evaluation problem and the application of urban sewage treatment via the application of public–private partnership models.

The foregoing studies provide evaluable references for the evaluation of industrial sewage treatment projects. However, the demand for industrial sewage treatment involves multiple goals. Furthermore, there are certain conflicts among different goals. In other words, it is difficult to develop a scheme whose indicators are all superior to those of others. Hence, a quantitative decision-making method for comprehensive evaluation is needed for the evaluation and selection of sewage treatment projects so as to improve the transparency of the evaluation process and lower subjectivity. On this basis, based on the foregoing relevant studies and the principles of systems science, this work constructed an evaluation indicator system for industrial sewage treatment projects, proposed a comprehensive evaluation method based on the entropy weight method and TOPSIS method and verified the feasibility and effectiveness of the evaluation system and method through a practical case.

## 3. Evaluation Indicator System for Industrial Sewage Treatment Projects

The establishment of an evaluation indicator system for industrial sewage treatment projects is critical to the comprehensive evaluation of sewage treatment projects. Whether such a system is reasonable or not will directly affect the correctness and comparability of the final evaluation results. The evaluation indicator system for industrial sewage treatment projects is a set of indicators used to forecast the comprehensive benefits of industrial sewage treatment projects. This set of indicators is established to reflect the effects of industrial sewage treatment projects on multiple aspects, i.e., the economy, the environment, technology, management and society, from different angles, levels and ranges based on the basic principles and rules of comprehensive evaluation of sewage treatment projects. However, with regard to comprehensive evaluation of industrial sewage treatment projects, due to the complexity of practical problems, some indicators may be not easily available or may demand high costs. Therefore, there is a restriction on the selection of indicators, although some of these indicators may also be relatively important to comprehensive evaluation of the projects. Due to differences among the projects, actually, there is no such authoritative method to judge an evaluation indicator system. Therefore, specific problems shall be dealt with carefully case by case. Specifically, for industrial sewage treatment projects, the aspects and factors involved are quite complicated. In this paper, the evaluation covers not only the environmental benefits brought by a reduction in sewage discharge but also some economic benefits. Furthermore, the implementation of such projects will also inevitably influence some aspects of society, such as improving residents' living standards, promoting regional development, etc. The evaluation of any benefit also involves many indicators, and its results are affected by a great deal of complicated factors. Generally, based on the specific situation of projects, appropriate indicators shall be selected as key factors among many influencing factors. These key factors constitute a multi-indicator system, which reflects a comprehensive evaluation system that contains most of the information that is needed for industrial sewage treatment project evaluation.

Based on the existing research, an on-site survey of several sewage treatment projects and the special characteristics of sewage treatment projects, we established a comprehensive evaluation indicator system for sewage treatment projects through an expert workshop in accordance with the basic rules of establishment of a comprehensive evaluation indicator system for projects. This indicator system mainly included 4 level-I indicators (environmental performance, economic performance, managerial performance and social performance) and 20 level-II indicators (Figure 1). Specifically, the indicator of environmental performance referred to the influences of the sewage treatment project's implementation on the natural environmental system and social environmental system, which included the subindicators of sewage treatment volume, compliance with sewage treatment

standards, reuse of sewage, secondary pollution and ecological and environmental coordination. Second, the indicator of economic performance referred to the economic costs and benefits of the sewage treatment project, and it included five level-II indicators, which were infrastructure costs, equipment costs, operational costs, direct benefits and indirect benefits. Third, the indicator of managerial performance referred to the performances of the process, operations and management of the sewage treatment project, which contained five subindicators of management safety, operational stability, management convenience, equipment integrity and equipment operational difficulty. Finally, the indicator of social performance referred to the social influences or contributions of the sewage treatment project, which included five level-II indicators of improvement of the regional investment environment, promotion of employment, improvement of residents' livelihoods, R&D of advanced technologies and process and role model effect.

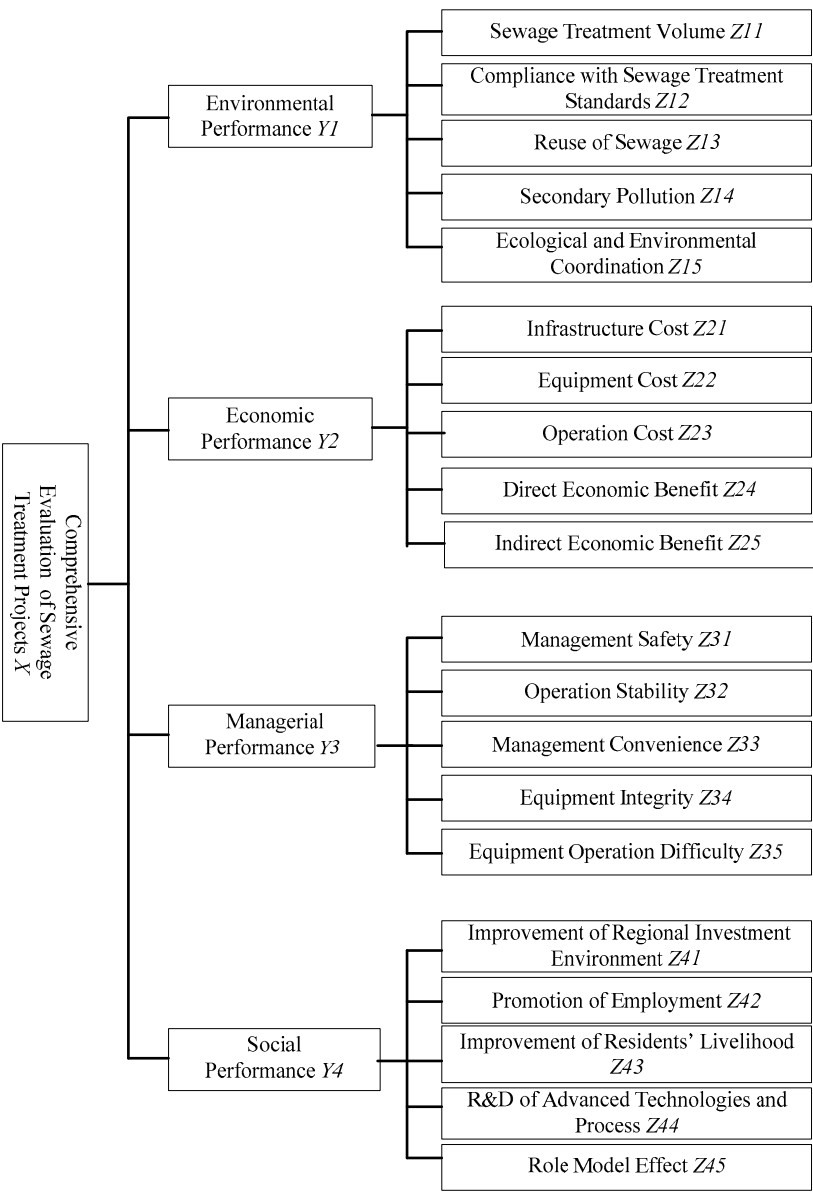

**Figure 1.** Evaluation indicator system for industrial sewage treatment projects.

## 4. Comprehensive Evaluation Method of Industrial Sewage Treatment Projects

Comprehensive evaluation of industrial sewage treatment projects is a complex systematic decision-making problem that involves multiple aspects of knowledge, such as environmental science,

economics, management science, sociology and systems science. The evaluation of sewage treatment projects shall target several goals, such as environmental friendliness, economic efficiency, management benefits and social benefits, etc. Therefore, it is a typical multitarget comprehensive evaluation. At present, many multitarget evaluation methods have been put forward [23–25]. However, when it comes to the evaluation of sewage treatment projects, which are characterized by high practicality and high complexity, the comments of decision makers and experts are often subjective and vague. As a result, the evaluation results vary greatly when different methods are adopted, surely bringing great difficulty to comprehensive evaluation of industrial sewage treatment projects [18].

Under the precondition that an evaluation indicator system for industrial sewage treatment projects has been established, the reliability and accuracy of evaluation results of sewage treatment projects are mainly dependent on two major factors: the determination of the weight of each evaluation indicator and the proposed comprehensive evaluation method. The traditional evaluation of sewage treatment projects is generally dependent on subjective judgments of decision makers and experts on the weight of each indicator. As individual experience and knowledge vary greatly, subjective elements are inevitably involved, causing great differences in the weight of the same evaluation indicator. As a result, the evaluation results may be greatly distorted or even cause mistakes in decision making [19,26]. In view of the foregoing problems, this work combined the subjective judgments of decision makers and experts with the objective situation of sewage treatment projects and applied an optimized entropy weight method to determine the weight of evaluation indicators for sewage treatment projects and correct the deviation of subjective judgments with a scientific weight coefficient; meanwhile, we adopted the improved TOPSIS as the comprehensive evaluation method for obtaining the most ideal comprehensive evaluation results.

### 4.1. The Entropy Weight Method and TOPSIS Method

The concept of entropy originates from the theory of thermodynamics. The value of entropy can serve as a measurement of the volume of valid information provided by a system, indicating the degree of randomness of the system. The entropy weight method is an evaluation method combining qualitative analysis with quantitative analysis, and it can determine the objective weights based on the variability of indicators. The entropy weight method is an important objective weighting method that has been widely applied in engineering technology, the social economy and other fields. In the entropy weight method, the weights of indicators are determined by the information content of each indicator transferring to decision makers. With regard to the evaluation of sewage treatment projects, suppose there are $m$ items to be evaluated and $n$ evaluation indicators. Then we can obtain the original evaluation matrix $E = (e_{ij})_{m \times n}$, where $e_{ij}$ means the value of the $j$th evaluation indicator of the $i$th item to be evaluated. Then, the value of the entropy of the indicator $e_j$ can be obtained using the following equation:

$$h_j = -\frac{1}{\ln m} \sum_{i=1}^{m} p_{ij} \ln p_{ij} \tag{1}$$

where $p_{ij} = \frac{e_{ij}}{\sum_{i=1}^{m} e_{ij}}$, $p_{ij}$ means the weight of the $i$th item to be evaluated under the $j$th indicator.

According to the definition and principle of entropy, the greater the value of the entropy of an indicator is, the less the valid information that will be provided by this indicator, and the smaller the role and weight of such an indicator in the comprehensive evaluation will be. Contrarily, the greater the value of the entropy is, the more the valid information provided by this indicator will be, and the greater the role and weight of such an indicator in the comprehensive evaluation will be [21].

The TOPSIS method, namely, the ideal point method, is a multitarget decision analysis method commonly used for limited solutions. With this method, the statistical data are converted into points in the multidimensional coordinate system, and the reference points (namely, the positive and negative ideal points) are identified in the space [27]. Then, the solutions are subjected to comprehensive ranking

based on their distances from the positive and negative ideal points, thus obtaining the comprehensive evaluation results. By using the TOPSIS method for comprehensive evaluation, this method normalized the decision matrix of the TOPSIS method and simplified the calculation of positive and negative ideal solutions to solve the complexity of the calculation of the Euclidean distance of evaluation objects from the positive and negative ideal points. At the same time, this paper introduced the concept of the relative proximity of the indicator value of each evaluation object to the ideal solution, and it arranged the order according to the relative proximity of each evaluation object [20].

### 4.2. Improved Entropy–TOPSIS Comprehensive Evaluation Method

The steps of the improved entropy–TOPSIS comprehensive evaluation method are shown as follows:

For the evaluation, which involves $m$ items to be evaluated and $n$ evaluation indicators, the original data matrix is $E = (e_{ij})_{m \times n}$.

$E$ matrix is nondimensionalized as $U = (u_{ij})_{m \times n}$, namely

$$u_{ij} = \frac{e_{ij}}{\left[ \sum\limits_{i=1}^{m} e_{ij}^2 \right]^{\frac{1}{2}}} \tag{2}$$

Calculate $p_{ij}$, namely, the weight of the $j$th indicator of the $i$th item to be evaluated.

$$p_{ij} = \frac{u_{ij}}{\sum\limits_{i=1}^{m} u_{ij}} \tag{3}$$

Calculate the entropy $h_j$ of the $j$th indicator.

$$h_j = -\frac{1}{\ln m} \sum\limits_{i=1}^{m} p_{ij} \ln p_{ij}, \ (j = 1, 2, 3 \ldots, n) \tag{4}$$

where $0 \leq h_j \leq 1$.

Calculate the diversity factor $g_j$ of the $j$th indicator.

$$g_j = 1 - h_j \tag{5}$$

For the $j$th indicator, the greater the value of $g_j$ is, the greater the role of the indicator in the evaluation of the scheme will be; contrarily, the smaller the value of $g_j$ is, the smaller the role of the indicator in the evaluation of the scheme will be.

Calculate the weight $w_j$ of the $j$th indicator.

$$w_j = \frac{g_j}{\sum\limits_{j=1}^{n} g_j} \tag{6}$$

Construct the weighted data matrix $R = (r_{ij})_{m \times n}$, where the element $r_{ij}$ can be obtained:

$$r_{ij} = w_j u_{ij} \tag{7}$$

Determine the positive ideal value $R^+$ and negative ideal value $R^-$ of the indicator. For the traditional TOPSIS method, due to the complexity of its positive and negative ideal values, it is difficult to calculate the Euclidean distance from each evaluation scheme to the positive and negative ideal points. Therefore, under the precondition that the evaluation results of the evaluation problems were

not affected, this paper made an improvement to simplify the calculation of the traditional TOPSIS method. In the normalized matrix $U$, the value of $u_{ij}$ is within $[0, 1]$. Here, let us set the preferable maximum target attribute value at $u_{ij} = 1$ and the preferable minimum target attribute value at $u_{ij} = 0$. We can know that if $r_j{}^+ = w_j$, $r_j{}^- = 0$, the positive and negative ideal solutions are

$$A^+ = (r_1{}^+, r_2{}^+ \ldots, r_j{}^+) = (w_1, w_2 \ldots, w_j)$$
$$A^- = (r_1{}^-, r_2{}^- \ldots, r_j{}^-) = (0, 0 \ldots, 0) \tag{8}$$

Calculate the Euclidean distance from each evaluation solution to the positive and negative ideal points. Euclid's formula is adopted for the calculation of such a distance.

$$D_i{}^- = \sqrt{\sum_{j=1}^{n} \left(r_{ij} - r_j{}^-\right)^2} = \sqrt{\sum_{j=1}^{n} w_j{}^2\left(u_{ij} - 0\right)^2}$$
$$D_i{}^+ = \sqrt{\sum_{j=1}^{n} \left(r_{ij} - r_j{}^+\right)^2} = \sqrt{\sum_{j=1}^{n} w_j{}^2\left(u_{ij} - 1\right)^2} \tag{9}$$

Calculate the relative proximity $c_i$ of the indicator value of each item to be evaluated to the ideal solution.

$$c_i = \frac{D_i{}^+}{D_i{}^+ + D_i{}^-} \tag{10}$$

The order of each project is arranged in accordance with the value of the relative proximity. The greater the relative proximity $c_i$ is, the better the result of comprehensive evaluation of the project will be; contrarily, the lower the value of $c_i$ is, the worse the comprehensive evaluation result of the project will be.

## 5. Case Study

This work took the industrial sewage treatment project of the China Water Affairs Group (hereinafter referred to as Q Company) in Q city, China, as the object of the case study so as to verify the feasibility and effectiveness of the proposed method in this paper. Q Company engages in investment, construction and operations management with regard to urban water supply and sewage treatment. Its business extends to comprehensive environmental governance, infrastructure consultation and construction and environmental protection technology services, etc., with a commitment to promoting the sustainable development of the social economy and environmental resources. Q Company takes up 97% and 94% of the market share in water supply and sewage treatment, respectively, in Q city. It runs 43 waterworks with a daily supply capacity of 3,360,000 cubic meters, serving a population of 8,120,000, and 77 sewage treatment works with a daily sewage treatment capacity of 3,700,000 cubic meters and a daily sludge treatment capacity of 800 tons.

In order to improve its industrial sewage treatment capacity and standards, Q Company has focused on the development of advanced oxidation technologies for industrial sewage treatment in the past three years. In the process of industrial production, there are many kinds of high-concentration organic pollutants and toxic pollutants in industrial sewage, and they are very harmful. The reaction process of the advanced oxidation technologies can generate a large number of extremely reactive free radicals (such as $\bullet OH$, etc.), thereby breaking up the refractory organic pollutants to form easily degradable and small molecules. For the advanced oxidation technology project, Q Company preliminarily proposed four alternative projects named A, B, C and D. Specifically, project A was mainly based on activated carbon technology, project B mainly focused on electrolytic oxidation technology, project C was mainly based on Fenton oxidation technology and project D mainly used ozone oxidation technology. To evaluate these four projects, Q Company organized a panel of experts in industrial sewage treatment to grade each item in the proposed evaluation indicator system. The range of scores

was 1–5. The higher the score was, the better the performance of the project on this indicator would be. Based on the grading results of the indicator system, Table 1 was obtained.

**Table 1.** The original data from the grading by the panel of experts.

| | $Z_{11}$ | $Z_{12}$ | $Z_{13}$ | $Z_{14}$ | $Z_{15}$ | $Z_{21}$ | $Z_{22}$ | $Z_{23}$ | $Z_{24}$ | $Z_{25}$ | $Z_{31}$ | $Z_{32}$ | $Z_{33}$ | $Z_{34}$ | $Z_{35}$ | $Z_{41}$ | $Z_{42}$ | $Z_{43}$ | $Z_{44}$ | $Z_{45}$ |
|---|---|---|---|---|---|---|---|---|---|---|---|---|---|---|---|---|---|---|---|---|
| A | 3 | 3 | 4 | 2 | 3 | 3 | 4 | 3 | 2 | 3 | 3 | 3 | 2 | 3 | 4 | 3 | 2 | 2 | 2 | 3 |
| B | 3 | 3 | 2 | 3 | 3 | 3 | 3 | 4 | 3 | 4 | 4 | 3 | 2 | 4 | 3 | 4 | 3 | 3 | 3 | 4 |
| C | 4 | 4 | 3 | 4 | 4 | 4 | 3 | 4 | 4 | 3 | 3 | 2 | 2 | 3 | 2 | 3 | 1 | 2 | 1 | 2 |
| D | 3 | 3 | 3 | 3 | 3 | 4 | 3 | 4 | 3 | 4 | 3 | 3 | 3 | 2 | 3 | 3 | 3 | 2 | 2 | 3 |

According to Formula (2), the original data from experts' grading score sheets were nondimensionalized and are shown in Table 2.

**Table 2.** The nondimensionalized data of experts' grading scores.

| | $Z_{11}$ | $Z_{12}$ | $Z_{13}$ | $Z_{14}$ | $Z_{15}$ | $Z_{21}$ | $Z_{22}$ | $Z_{23}$ | $Z_{24}$ | $Z_{25}$ | $Z_{31}$ | $Z_{32}$ | $Z_{33}$ | $Z_{34}$ | $Z_{35}$ | $Z_{41}$ | $Z_{42}$ | $Z_{43}$ | $Z_{44}$ | $Z_{45}$ |
|---|---|---|---|---|---|---|---|---|---|---|---|---|---|---|---|---|---|---|---|---|
| A | 0.457 | 0.457 | 0.649 | 0.324 | 0.457 | 0.424 | 0.610 | 0.397 | 0.324 | 0.424 | 0.457 | 0.539 | 0.436 | 0.487 | 0.649 | 0.457 | 0.417 | 0.436 | 0.471 | 0.487 |
| B | 0.457 | 0.457 | 0.324 | 0.487 | 0.457 | 0.424 | 0.457 | 0.530 | 0.487 | 0.566 | 0.610 | 0.539 | 0.436 | 0.649 | 0.487 | 0.610 | 0.626 | 0.655 | 0.707 | 0.649 |
| C | 0.610 | 0.610 | 0.487 | 0.649 | 0.610 | 0.566 | 0.457 | 0.530 | 0.649 | 0.424 | 0.457 | 0.359 | 0.436 | 0.487 | 0.324 | 0.457 | 0.209 | 0.436 | 0.236 | 0.324 |
| D | 0.457 | 0.457 | 0.487 | 0.487 | 0.457 | 0.566 | 0.457 | 0.530 | 0.487 | 0.566 | 0.457 | 0.539 | 0.655 | 0.324 | 0.487 | 0.457 | 0.626 | 0.436 | 0.471 | 0.487 |

The weight $p_{ij}$ of the *i*th indicator of the *j*th evaluation item was calculated in accordance with Formula (3). Then, the weight $P_{ij}$ was obtained, as shown in Table 3.

**Table 3.** The weight of the *j*th indicator in the *i*th scheme.

| | $Z_{11}$ | $Z_{12}$ | $Z_{13}$ | $Z_{14}$ | $Z_{15}$ | $Z_{21}$ | $Z_{22}$ | $Z_{23}$ | $Z_{24}$ | $Z_{25}$ | $Z_{31}$ | $Z_{32}$ | $Z_{33}$ | $Z_{34}$ | $Z_{35}$ | $Z_{41}$ | $Z_{42}$ | $Z_{43}$ | $Z_{44}$ | $Z_{45}$ |
|---|---|---|---|---|---|---|---|---|---|---|---|---|---|---|---|---|---|---|---|---|
| A | 0.231 | 0.231 | 0.333 | 0.167 | 0.231 | 0.214 | 0.308 | 0.200 | 0.167 | 0.214 | 0.231 | 0.273 | 0.222 | 0.250 | 0.333 | 0.231 | 0.222 | 0.222 | 0.250 | 0.250 |
| B | 0.231 | 0.231 | 0.167 | 0.250 | 0.231 | 0.214 | 0.231 | 0.267 | 0.250 | 0.286 | 0.308 | 0.273 | 0.222 | 0.333 | 0.250 | 0.308 | 0.333 | 0.333 | 0.375 | 0.333 |
| C | 0.308 | 0.308 | 0.250 | 0.333 | 0.308 | 0.286 | 0.231 | 0.267 | 0.333 | 0.214 | 0.231 | 0.182 | 0.222 | 0.250 | 0.167 | 0.231 | 0.111 | 0.222 | 0.125 | 0.167 |
| D | 0.231 | 0.231 | 0.250 | 0.250 | 0.231 | 0.286 | 0.231 | 0.267 | 0.250 | 0.286 | 0.231 | 0.273 | 0.333 | 0.167 | 0.250 | 0.231 | 0.333 | 0.222 | 0.250 | 0.250 |

The entropy, variable coefficient and weight of each indictor were calculated in accordance with Formulas (4)–(6), as shown in Table 4.

**Table 4.** The entropy, variable coefficient and weight of each indicator.

| | $Z_{11}$ | $Z_{12}$ | $Z_{13}$ | $Z_{14}$ | $Z_{15}$ | $Z_{21}$ | $Z_{22}$ | $Z_{23}$ | $Z_{24}$ | $Z_{25}$ | $Z_{31}$ | $Z_{32}$ | $Z_{33}$ | $Z_{34}$ | $Z_{35}$ | $Z_{41}$ | $Z_{42}$ | $Z_{43}$ | $Z_{44}$ | $Z_{45}$ |
|---|---|---|---|---|---|---|---|---|---|---|---|---|---|---|---|---|---|---|---|---|
| $h_j$ | 0.994 | 0.994 | 0.980 | 0.980 | 0.994 | 0.993 | 0.994 | 0.995 | 0.980 | 0.993 | 0.994 | 0.990 | 0.987 | 0.980 | 0.980 | 0.994 | 0.946 | 0.987 | 0.953 | 0.980 |
| $g_j$ | 0.006 | 0.006 | 0.020 | 0.020 | 0.006 | 0.007 | 0.006 | 0.005 | 0.020 | 0.007 | 0.006 | 0.010 | 0.013 | 0.020 | 0.020 | 0.006 | 0.054 | 0.013 | 0.047 | 0.020 |
| $w_j$ | 0.019 | 0.019 | 0.065 | 0.065 | 0.019 | 0.023 | 0.019 | 0.016 | 0.065 | 0.023 | 0.019 | 0.030 | 0.040 | 0.065 | 0.065 | 0.019 | 0.173 | 0.040 | 0.150 | 0.065 |

The weighted normalized data matrix could be obtained using Formula (7), as shown in Table 5.

**Table 5.** The weighted normalized data matrix.

| | $Z_{11}$ | $Z_{12}$ | $Z_{13}$ | $Z_{14}$ | $Z_{15}$ | $Z_{21}$ | $Z_{22}$ | $Z_{23}$ | $Z_{24}$ | $Z_{25}$ | $Z_{31}$ | $Z_{32}$ | $Z_{33}$ | $Z_{34}$ | $Z_{35}$ | $Z_{41}$ | $Z_{42}$ | $Z_{43}$ | $Z_{44}$ | $Z_{45}$ |
|---|---|---|---|---|---|---|---|---|---|---|---|---|---|---|---|---|---|---|---|---|
| A | 0.009 | 0.009 | 0.042 | 0.021 | 0.009 | 0.010 | 0.012 | 0.006 | 0.021 | 0.010 | 0.009 | 0.016 | 0.017 | 0.032 | 0.042 | 0.009 | 0.072 | 0.017 | 0.071 | 0.032 |
| B | 0.009 | 0.009 | 0.021 | 0.032 | 0.009 | 0.010 | 0.009 | 0.008 | 0.032 | 0.013 | 0.012 | 0.016 | 0.017 | 0.042 | 0.032 | 0.012 | 0.108 | 0.026 | 0.106 | 0.042 |
| C | 0.012 | 0.012 | 0.032 | 0.042 | 0.012 | 0.013 | 0.009 | 0.008 | 0.042 | 0.010 | 0.009 | 0.011 | 0.017 | 0.032 | 0.021 | 0.009 | 0.036 | 0.017 | 0.035 | 0.021 |
| D | 0.009 | 0.009 | 0.032 | 0.032 | 0.009 | 0.013 | 0.009 | 0.008 | 0.032 | 0.013 | 0.009 | 0.016 | 0.026 | 0.021 | 0.032 | 0.009 | 0.108 | 0.017 | 0.071 | 0.032 |

The Euclidean distance from each evaluation solution to the positive and negative ideal points, $D_i^+$ and $D_i^-$, was obtained using Formulas (8) and (9), as shown in Table 6.

**Table 6.** The Euclidean distance $D_i^+$ and $D_i^-$.

| Euclidean Distance | A | B | C | D |
|---|---|---|---|---|
| $D_i^+$ | 0.215 | 0.183 | 0.240 | 0.201 |
| $D_i^-$ | 0.135 | 0.179 | 0.103 | 0.156 |

Finally, using the improved entropy–TOPSIS comprehensive evaluation method, the relative proximity $c_i$ and its order could be obtained using Formula (10), as shown in Table 7.

**Table 7.** The relative proximity $c_i$ and its order via the improved entropy–TOPSIS method.

| Evaluation Item | A | B | C | D |
|---|---|---|---|---|
| $c_i$ | 0.615 | 0.506 | 0.699 | 0.563 |
| Order | 2 | 4 | 1 | 3 |

As seen in Table 7, the comprehensive evaluation results of Q Company's four sewage treatment projects were C⟩A⟩D⟩B; thus, project C, that is, the Fenton oxidation technology project, was ranked as the best project in this evaluation. The results were consistent with those obtained by the unsimplified traditional TOPSIS method (Table 8). Throughout the evaluation process of the improved entropy–TOPSIS method and traditional TOPSIS method, the improved entropy–TOPSIS method could obtain results consistent with those of the traditional TOPSIS method. Moreover, the improved entropy–TOPSIS method was more concise and efficient than the traditional TOPSIS method because it avoided the step of determining and calculating the specific positive and negative ideal solutions so as to effectively simplify the traditional TOPSIS method. Additionally, in the improved entropy–TOPSIS method, its simplified setting of ideal values could help decision makers easily understand and run the method. To sum up, the improved entropy–TOPSIS method had higher reliability and practicability with its clear idea and easy calculations.

**Table 8.** The relative proximity $c_i$ and its order via the traditional TOPSIS method.

| Evaluation Item | A | B | C | D |
|---|---|---|---|---|
| $c_i$ | 0.502 | 0.337 | 0.649 | 0.439 |
| Order | 2 | 4 | 1 | 3 |

## 6. Conclusions

Focusing on the comprehensive evaluation of industrial sewage treatment projects, this paper constructed an evaluation indicator system that was able to thoroughly reflect the environmental performance, economic performance, managerial performance and social performance of industrial sewage treatment projects. Practice proved that this indicator system was able to evaluate the performance of sewage treatment projects at various levels in a scientific, systematic and comprehensive manner. This work used the entropy method to make a modification to the experience-based grading of experts and thus determined the weight of each evaluation indicator of sewage treatment projects. In this way, the subjectivity in traditional experts' evaluations and other multilevel and multi-indicator weight calculation methods was avoided, thus making the evaluation results more objective, more accurate and more practical. Furthermore, the evaluation results with the improved TOPSIS method and traditional TOPSIS method were consistent, and moreover, the calculation process of the improved TOPSIS method was more concise, more efficient and clearer than that of the traditional TOPSIS method. In conclusion, the evaluation system and evaluation method proposed in this paper for sewage treatment projects are reasonable and practical to some extent. They will help with the objective, overall and systematic evaluation of industrial sewage treatment projects and provide preference

evaluation and decision-making support for techniques demonstrating and projects promoting industrial sewage treatment.

**Author Contributions:** Conceptualization, X.Y. and J.S.; methodology, X.Y. and S.S.; formal analysis, X.Y. and S.S.; supervision, J.S.; writing—original draft preparation, X.Y. and J.S.; writing—review and editing, X.Y., S.S. and J.S. All authors have read and agreed to the published version of the manuscript.

**Funding:** This work was supported by the National Key Research and Development Program of China (2016YFE0205600).

**Conflicts of Interest:** The authors declare no conflict of interest.

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
