# Peer review of "A Comprehensive Evaluation Method for Industrial Sewage Treatment Projects Based on the Improved Entropy-TOPSIS"

_sustainability, doi:10.3390/su12176734_

Round 1

Reviewer 1 Report

The indicator system are questionable. It is not a problem in Topsis if some indicators are correlated. However it seems to me the intersections of the 4 first level indicators are too large and some sub-indicators are classified, ex. stimulation of economic growth Z41 belongs to economic benefit. This may lead to non-negligible bias in the result. Personally, I suggest the authors revise and modify their indicator definitions. 

Author Response

Respond to the comments of reviewer 1

Point: The indicator system are questionable. It is not a problem in Topsis if some indicators are correlated. However it seems to me the intersections of the 4 first level indicators are too large and some sub-indicators are classified, ex. stimulation of economic growth Z41 belongs to economic benefit. This may lead to non-negligible bias in the result. Personally, I suggest the authors revise and modify their indicator definitions.

Respond: Thanks for the valuable comment and suggestion. According to this valuable suggestion, we have carefully revised the evaluation indicator system to better differentiate the level I and level II indicators. Moreover, we have further revised and modified the indicators’s definitions and expressions to avoid the ambiguity among them. As for the stimulation of economic growth Z41, it does have some ambiguity. We have further consulted the experts and thought deeply about this indicator, then we revised it as “improvement of regional investment environment Z41”, which is a more precise expression and defination.

Reviewer 2 Report

Manuscript Number:  

Authors: Xin Yu and Jiafu Su

Title: A Comprehensive Evaluation Method for Industrial Sewage Treatment Projects Based on the Improved Entropy-TOPSIS

Journal:  Sustainability

The manuscript addresses an improvement on the traditional TOPSIS method and simplified

the calculation about TOPSIS-based comprehensive evaluation. A case study is presented

applying the methodology by evaluating sewage treatment projects of the Water Affairs Group Company in Q city, China.

General Comments

In my opinion, the manuscript should have minor revision before the final acceptance.

The manuscript is using standard English. The authors should apply the standard TOPSIS to show the difference in somehow to the proposed improved methodology. What are the advantages of the improved method compared with the classic method? In the final result or in time of application? Or another reason.

About the case study it will be interesting to describe better what are the treatment projects A, B,C and D. Without those descriptions it is equivalent to a generic application without “face”.

Specific Comments:

  • line 59 – Confirm that the sewage treatment is for the industry. Is this applicable to the domestic effluents with more impact ( in global analysis).
  • line 76 – it is mention that this is applied to domestic sewage. Please confirm the sentence.
  • line 81- please explain what is “DEA”
  • Line 158- improve the sentence “It is an objective weighting method.”
  • Line 160- “transferred to from decision makers.”
  • Eq 9 and 10 the wij should be replaced by wj
  • The values in table 4 should be confirmed, I got different values
  • Please show the Di+ and Di- values.
  • In my calculations, the values of ci are different from the ones in table 5
  • I calculated the average of Z values in table 1 for each alternative i, and I obtained the highest value for the alternative B, as the conclusion of your study after the application of the proposed method. Could you comment on this?

Author Response

Respond to the comments of reviewer 2

Point 1: The manuscript addresses an improvement on the traditional TOPSIS method and simplified the calculation about TOPSIS-based comprehensive evaluation. A case study is presented applying the methodology by evaluating sewage treatment projects of the Water Affairs Group Company in Q city, China.

In my opinion, the manuscript should have minor revision before the final acceptance.

Respond 1: Thanks for the valuable comment and suggestion. Your valuable suggestion are very helpful to improve our work. According to the suggestions, we have carefully revised the paper. The detailed responds to the comments are shown as follows.

Point 2: The manuscript is using standard English. The authors should apply the standard TOPSIS to show the difference in somehow to the proposed improved methodology. What are the advantages of the improved method compared with the classic method? In the final result or in time of application? Or another reason.

Respond 2: Thanks for the valuable comment and suggestion. According to the suggestions, we have add the evaluation results of the standard TOPSIS method, which is are consistent with those obtained by the improved Entropy-TOPSIS method proposed in this paper. Moreover we have further discussed the advantages of the improved Entropy-TOPSIS method compared with the standard TOPSIS method, which is “the improved Entropy-TOPSIS method is more concise and efficient than the traditional TOPSIS method, because it avoids the step to determine and calculate the specific positive and negative ideal solutions, so as to effectively simplifying the traditional TOPSIS method. Addtitionally, in the improved Entropy-TOPSIS method, its simplified setting of ideal values can help decision maker easily understand and run the method.” The detailed revision can be seen in the paragraph underTable 7.

Point 3: About the case study it will be interesting to describe better what are the treatment projects A, B,C and D. Without those descriptions it is equivalent to a generic application without “face”.

Respond 3: Thanks great for the valuable suggestion. According to the suggestion, we have further describe and highlight the information about the four sewage treatment projects to give readers a better undrstanding about this work. The detailed descriptions about the four project are “In order to improve its industrial sewage treatment capacity and standard, Q Company focuses on the development of the advanced oxidation technologies for industrial sewage treatment in recent three years. In the process of industrial production, there are many kinds of high-concentration organic pollutants and toxic pollutants in the industrial sewerage, and they are very harmful. The reaction process of the advanced oxidation technologies can generate a large number of extremely reactive free radicals (such as •OH, etc.), thereby breaking up the refractory organic pollutants to form easily degradable and degradable small molecules. For the dvanced oxidation technology project, Q Company tpreliminarily proposed four alternative projects named A, B, C and D. Specifically, project A is mainly based on the activated carbon technology, project B mainly fouces on the electrolytic oxidation technology, project C is mainly based on the Fenton oxidation technology, and project D mainly uses the ozone oxidation technology.” The detailed revision can be seen in the paragraph underTable 7.

Point 4: line 59 – Confirm that the sewage treatment is for the industry. Is this applicable to the domestic effluents with more impact ( in global analysis).

Respond 4: Thanks great for the valuable comment. This work mainly focuses on the industrial sewage treatment project. Its research ideas and results can provide a helpful reference for the domestic effluents treatment mamagement. For the sentence of line 59, it is a little unclear, so we have further revised it to aviod the misunderstanding. The detailed revision can be seen in the line 59-64.

Point 5: line 76 – it is mention that this is applied to domestic sewage. Please confirm the sentence.

line 81- please explain what is “DEA”

Line 158- improve the sentence “It is an objective weighting method.”

Line 160- “transferred to from decision makers.”

Eq 9 and 10 the wij should be replaced by wj

Respond 5: Thank you very much for your careful suggestions. We are sorry for these errors, and we have carefully revised them according to these suggestions.

Moreover, we have carefully checked the whole paper and revised the similar errors. The detailed revisions can be seen in the revised article.

Point 6: The values in table 4 should be confirmed, I got different values

Please show the Di+ and Di- values.

In my calculations, the values of ci are different from the ones in table 5

Respond 6: Thanks great for your careful suggestions. Sorry for this issue, we have carefully checked the data and calculation process, and I found some mistakes in the original data, and an error in the calculation step of original data nondimensionalization. Then, we have carefully revised thees errors and recalculate the results. In addition, we have added the Di+ and Di- values in Table 6. Thanks again for your suggestions. The detailed revisions can be seen in the Section 5.

Point 7: I calculated the average of Z values in table 1 for each alternative i, and I obtained the highest value for the alternative B, as the conclusion of your study after the application of the proposed method. Could you comment on this?

Respond 7: Thanks great for your careful comments. For the original version of the paper, it really has this same. However, it is a coincidence, because the average values of original data dose not take into account the different weights of indicatiors. In the revised manuscript, after revising the errors of original data and calculation process, the best solution is alternative C, which does have this phenomenon.

Round 2

Reviewer 1 Report

The revised version is much better.